# Correlation of clinical decision-making with probability of disease: A web-based study among general practitioners

Lionel De Alencastro[1]*, Isabella Locatelli[1], Carole Clair[1], Mark H. Ebell[2], Nicolas Senn[1]

1 Center for Primary Care and Public Health (Unisanté), Lausanne, Switzerland, 2 Department of Epidemiology and Biostatistics, College of Public Health, The University of Georgia, Athens, Georgia, United States of America

* Lionel.De-Alencastro@unisante.ch

## Abstract

### Background

Medical decision-making relies partly on the probability of disease. Current recommendations for the management of common diseases are based increasingly on scores that use arbitrary probability thresholds.

### Objective

To assess decision-making in pharyngitis and appendicitis using a set of clinical vignettes, and the extent to which management is congruent with the true probability of having the disease.

### Design

We developed twenty-four clinical vignettes with clinical presentations corresponding to specific probabilities of having disease defined by McIsaac (pharyngitis) or Alvarado (appendicitis) scores. Each participant answered four randomly selected web-based vignettes.

### Participants

General practitioners (GP) working in primary care structures in Switzerland and the USA.

### Main measures

A comparison between the GP's management decision according to the true probability of having the disease and to the GP's estimated probability, investigating the GP's ability to estimate probability of disease.

### Key results

The mean age of the GPs was 48 years (SD 12) and 66% were men. The correlation between the GP's clinical management decision based on the vignette and the

**Data Availability Statement:** The authors have uploaded the dataset on Zenodo: Lionel De Alencastro, Isabella Locatelli, Carole Clair, Mark H. Ebell, & Nicolas Senn. (2020). Dataset of

Correlation of clinical decision-making with probability of disease: A web-based study among general practitioners [Data set]. Zenodo. http://doi.org/10.5281/zenodo.4094682.

**Funding:** The authors received no specific funding for this work.

**Competing interests:** The authors have declared that no competing interests exist.

recommendations was stronger for appendicitis than pharyngitis ($k_w$ = 0.74, 95% CI 0.70–0.78 vs. $k_w$ = 0.66, 95% CI 0.62–0.71). On the other hand, the association between the clinical management decision and the probability of disease estimated by GPs was more congruent with recommendations for pharyngitis than appendicitis ($k_w$ = 0.70, 95% CI 0.66–0.73 vs. 0.61, 95% CI 0.56–0.66). Only a minority of GPs correctly estimated the probability of disease (29% for appendicitis and 39% for pharyngitis).

## Conclusions

Despite the fact that general practitioners often misestimate the probability of disease, their management decisions are usually in line with recommendations. This means that they use other approaches, perhaps more subjective, to make decisions, such as clinical judgment or reasoning that integrate factors other than just the risk of the disease.

## Introduction

Medical decision-making relies, at least partly, on the probability of disease, which helps define appropriate thresholds to treat or to do further investigation. According to the likelihood of the disease, three management options usually exist: the diagnosis is excluded, the diagnosis is uncertain and more investigation is needed, or the disease is likely enough to initiate treatment without further investigation. This was described as the "decision threshold model" of disease by Pauker et al in 1980 [1]. The test threshold is the probability below which no action (testing or treating) is necessary because the likelihood of a disease is too low to warrant further testing. The treatment threshold is the point above which confidence in the diagnosis is high enough that therapy can be initiated without testing.

Current recommendations for the management of common and important conditions, like deep venous thrombosis (Wells score [2, 3]) or pulmonary embolism (Wells/Geneva score [4]) are increasingly based on clinical scores that define thresholds based on probability. With the emergence of evidence-based medicine, clinical epidemiology data reinforces the general practitioner's (GP's) experience with the potential to improve clinical practice, which is subjective and sometimes biased [5]. The objectives of this clinical prediction rule-centered strategy are to avoid unnecessary investigation (often expensive and sometimes harmful) or overtreatment (leading to antibiotic resistance or side effects) and to minimize misdiagnosis.

Although allocating patients to one of the three management categories according to probability threshold (low [diagnosis excluded], moderate [more investigation] and high-risk [initiate treatment] likelihood) facilitates patient management, it should be kept in mind that the decision thresholds are somewhat arbitrarily set. Indeed, they result from a combination of test performance analysis and expert opinion on the acceptable risk of having the disease in question. They are therefore theoretically set and based on probability approach and are not necessarily concordant with the physician's judgment of the clinical decision.

Very few studies in the literature have explored a link between decision-making by GPs and the probability of disease. In 1983, Eisenberg and Hershey presented a clinical vignette to GPs to investigate their decision-making regarding testing, treatment or neither and determined a plausible range for the GP's test and treatment thresholds [6]. In 2015, Ebell and colleagues used clinical vignettes to calculate test and treatment thresholds for several common diseases

[7]. They also compared Switzerland to the USA and found that test and treatment thresholds were different between the two countries.

In a further study in 2018, Ebell and colleagues explored decision thresholds in community-acquired pneumonia (CAP) [8]. Results showed the GPs' difficulty in estimating the likelihood of having disease in that >82% tended to overestimate the risk of having CAP. This suggests that clinical decision-making is probably disconnected from the true likelihood of disease and is rather based on the GP's self-estimated probability of having disease. This then challenges how thresholds are set in scores and what can be considered as an "acceptable" probability to rule in or out a disease.

In light of these studies, the present vignette–based study aims to compare two approaches; first, the GP's subjective assessment of a clinical situation, to second, the objective data related to the recommendations and to see if they are congruent.

The first approach is the GP who decides on the management of a patient based on his own (subjective) estimation of the probability of having the disease, linked to the general context that integrates other factors (such as his relation with the patient or the risks incurred in case of medical error for example). The second, more objective approach, corresponds to the true probability of having the disease in a given situation, that can be explored through clinical scores (Fig 1).

In evidence-based medicine, the use of tools such as clinical scores has become an important part of clinical reasoning. Scores are more and more numerous and widely used as many kinds of health applications tools are available on-line (internet, mobile phones). Given the somewhat arbitrary nature of these tools, it is necessary to assess whether the recommendations are in line with the clinical practice. If, in a given clinical situation, the management by GPs differs greatly from the recommended guidelines, we may wonder if the development of the clinical score corresponds to clinical management. It is also important for GPs to know whether their clinical sense allows them to deliver clinical management that conforms to guidelines, and if not, understand the potential factors that prevent them from doing so.

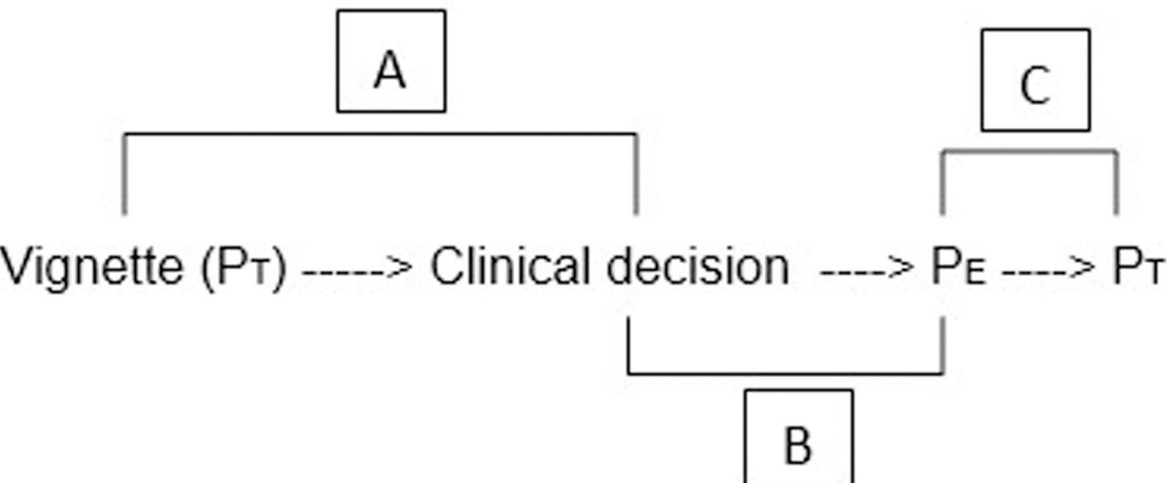

**Fig 1. Path through the questionnaire.** $P_T$: True probability, $P_E$: Estimated probability. A: compares the clinical decisions of the GPs to the true probabilities of the vignettes ($P_T$). B: compares the clinical decisions of the GPs to their estimation of the probabilities ($P_E$). C: compares the estimated probability ($P_E$) of the GPs to the true probability ($P_T$) of the vignette.

## Methods

This is an observational study using clinical vignettes presented to GPs concerning pharyngitis and appendicitis scenarios. The choice of the two common diseases was based on their gender neutrality, their frequency of presentation in the ambulatory-care setting and the availability of internationally-validated evidence-based practice recommendations using clinical scores [9, 10].

Acute abdominal pain accounts for about 1–2% of the reasons for consulting primary care physicians and appendicitis is often require exclusion as part of the differential diagnosis [11]. A previous American study calculated a cumulative lifetime risk to have an appendicitis of 8.6% for males and 6.7% for females which is significant [12]. Common signs and symptoms are part of the Alvarado score (migration of pain, anorexia, nausea, tenderness in the right lower quadrant, rebound pain, elevated temperature, leukocytosis, left shift of the white blood cell count). After the clinical examination and realization of a blood test in some cases, GPs usually organize a radiological exam (US/CT) if readily available or may directly address the patient to an emergency service. Concerning pharyngitis, a previous National Ambulatory Medical Care Survey in 1995 showed that it was the third leading cause of medical practice visits, accounting for 4.3% of visits [13]. It is important to exclude streptococcal pharyngitis as it is an indication for antibiotics, unlike the case of a viral infection. Common signs and symptoms are part of the McIsaac score (temperature over 38˚C, no cough, tender anterior cervical adenopathy, tonsillar swelling or exudate) but in addition to odynophagia, malaise, headache, moderate neck stiffness and digestive disorders (inappetence, nausea, vomiting and abdominal pain) may be present. Throat cultures and rapid streptococcal antigen tests are usually used for the diagnosis. Until recently, antibiotic treatment was indicated in most cases.

Twelve different vignettes (6 for pharyngitis and 6 for appendicitis) were created, each with a specific probability range of having the disease from a combination of different signs and symptoms included in the McIsaac Score [14] (pharyngitis) or Alvarado Score [10] (appendicitis). Studies exploring the probability of having the disease for each individual value of the score were used to set the true vignettes probability (S1 Table). This is indeed the only way to generate vignettes with specific clinical patterns that match specific probabilities (or a narrow range of probabilities).

The vignettes were stratified by risk category as in the scores: for appendicitis: "low risk" as Alvarado score 1 (<39% probability to have appendicitis) or 4 (47–58%); "moderate risk" as Alvarado score 5 (50–64%) or 6 (57–71%); and "high risk" as Alvarado score 7 (61–79%) or 9 (66–99%). For pharyngitis: "low risk" as McIsaac score 0 (1.0–2.5% probability of streptococcal pharyngitis) or 1 (5–10%); "moderate risk" as McIsaac score 2 (11–17%) or 3 (28–35%); and "high risk" as McIsaac score 4 (51–53%), or 5 (>53%). This is summarized in S1 Table, and an example of a vignette is available in the S1 Text.

In addition, each vignette had two versions, one with a female patient and the other with a male patient. Thus, in total there were 24 clinical vignettes randomly split into six different questionnaires. We used the SurveyMonkey® software to produce the questionnaires, integrating four vignettes per questionnaire. Several possible management options were proposed in each vignette. In addition, physicians had the possibility of proposing another option using free text (S2 Table).

### Setting

Over a period of 15 months (from May 2015 to July 2016), we contacted GPs in the USA and Switzerland and collected data. As clinical practices and guidelines may vary from place to place, it is interesting to study two different contexts. Moreover, comparing two countries

strengthens the external validity and generalizability of the results. Indeed, in the secondary analyses, it allowed us to see differences between the two. In Switzerland, we emailed GPs who were members of the Sentinella and SPAM (Swiss Primary Care Active Monitoring) networks. Sentinella is an epidemiological surveillance-system of the Federal Office of Public Health (FOPH) of Swiss physicians who report morbidity data on communicable diseases weekly. SPAM is a network of Swiss GPs created to monitor the primary care system. The members of these networks are spread all over the country. We randomly assigned one of the six questionnaires to each email address. GPs working at the Center for Primary Care and Public Health (Unisanté) in Lausanne were also contacted by email. Finally, we randomly distributed flyers at a meeting for continuous medical education at the University Hospital (Centre Hospitalier Universitaire Vaudois) of Lausanne. For the USA, we inserted a banner on the Daily POEMs website (http://www.essentialevidenceplus.com) that is read around the world, but mainly in the USA and Canada. The physicians who clicked on the banner were randomly allocated to one of the six online questionnaires.

The eligibility criteria were GPs working in a primary care structure (private practice or ambulatory medical center) in Switzerland or the USA. We aimed for a sample size of 200 participants for each country. This study did not require ethical clearance under Swiss regulation as no biomedical information was collected.

## Analyses

First, we assessed how the GP's management decision correlated with the recommendations based on the undisclosed true probability of disease based on S1 Table (Fig 1A). GPs were asked to estimate the probability of the disease based on the clinical description in the vignette. It was then possible to assess how the GP's decision corresponded to their personal estimate of probability (Fig 1B). We then compared the true and estimated probabilities to see how accurately GPs could estimate probability of disease (Fig 1C). After completion of the questionnaire, we revealed the true probability to the GPs. Finally, in order to identify potential factors influencing decision making, we collected GP demographic characteristics such as gender, age, country of residence and the availability of ultrasound (US) or computed tomography (CT) in their health care center.

All online questionnaires were filled in anonymously and attached to an IP address to allow only a single use. After answering a question, it was not possible to go back to the previous page to avoid making corrections according to the results. The questionnaire was available in English, French and German.

## Statistical methods

Physician characteristics were summarized descriptively for the whole sample and stratified by country. Agreement between the physician's management decision (symptomatic treatment only, further investigation, or initiate treatment) and the one given by the true probability of disease (low = > symptomatic, medium = >investigation, high = > treatment) was evaluated by means of a weighted kappa ($k_w$) with quadratic weights. The latter gives larger weight to wider disagreement (symptomatic/treatment) than to disagreements being less far apart on the ordinal scale (symptomatic/investigation or investigation/treatment). Index $k_w$ takes values between zero and one; zero representing agreement uniquely due to chance and one the maximum agreement. The same procedure was adopted to evaluate agreement between physician management decision and the management suggested by their estimated probability of disease. The latter was defined as "low" (suggested symptomatic) if <0.5 for appendicitis and <0.1 for

pharyngitis, as "high" (suggested treatment) if >0.7 for appendicitis and >0.4 for pharyngitis and as "moderate" (suggested investigation) in the intervals between these cut-offs.

In order to compare the true probability with the estimated probability of disease, we attributed the probability estimated by the physician to the risk score interval whose center had the smallest distance from the estimated probability itself and calculated for each disease, the percentage probability of under/over/correct estimation by physicians.

All comparisons were stratified for physician/patient characteristics: sex of physician, country, age of physician (cut-off 50 years, approximately the median age) and sex of the patient.

Test and treatment thresholds were determined using the method described by Ebell and colleagues. This method is based on a logistic regression analysis of the physician decision with respect to the true probability of the disease:

$$\ln[\mathrm{p}/(1-\mathrm{p})] = a + bx \qquad (1)$$

where $p$ is the probability of not ruling out when the test threshold is determined and the probability of treating when the treatment threshold is determined; $x$ is the true disease probability defined as the midpoint of each score probability interval, and $a$ and $b$ are regression coefficients. The test (respectively, treatment) threshold is defined as the disease probability such that the corresponding probability of not ruling out (respectively, treating) is equal to 0.5. Considering that in our study each physician evaluated several vignettes, we adopted generalized estimating equations (GEE). The latter generalizes the logistic model (1) allowing for correlation between decisions of a same physician faced with different vignettes (unstructured correlation option).

The model was adjusted in turn for sex of physician, country (USA vs Switzerland), age ($\leq$50 vs >50) and sex of patient, allowing a statistical comparison between test and treatment thresholds for subgroups of populations defined by each of these dichotomous variables.

We performed statistical analyses using the R software package (R Core Team [2013]. R: A language and environment for statistical computing. R Foundation for Statistical Computing, Vienna, Austria. http://www.R-project.org/)

## Results

In Switzerland, 201 of 350 contacted GPs answered the online questionnaire (response rate = 57%), of which, 194 completed the entire questionnaire (97%), 7 incompletely. For the daily POEMs link, 128 answers were registered: 86 from Americans; 13 from other countries (Italy, Japan, Austria, Canada, Saudi Arabia, Brazil, New Zealand and Germany) and 29 incompletely filled in (22%). We collected all available questionnaire data (even incompletely filled in). Overall, participants responded to 1'248 clinical vignettes: 631 for pharyngitis (317 male patients, 314 females) and 617 for appendicitis (309 male patients, 308 females). We compared USA to Switzerland (all questionnaires from other countries were excluded), using the 287 questionnaires available for analysis.

The majority of responding GPs were male (66%). The mean age was 48 years (SD 12) and 78% were working in private practice. 40% of the GPs had the possibility to do US or CT scans in the same building. The participant's characteristics are summarized in Table 1.

### Main results

Agreement between the physician management decision (symptomatic treatment only, further investigation or initiate treatment) and the true probability of disease (low, medium, high) was higher for appendicitis than for pharyngitis ($k_w$ = 0.74, 95% CI 0.70–0.78 versus $k_w$ = 0.66, 95% CI 0.62–0.71; Fig 1A and Table 2). Stratifying for individual physician/patient

**Table 1. Baseline characteristics of the participants.**

|  | **Total** | **Switzerland** | **USA** |
|---|---|---|---|
| Women, N (%) [*] | 95 (33.93%) | 67 (34.54%) | 28 (32.56%) |
| Age, mean (SD) [*] | 48.86 (12.06) | 50.20 (11.98) | 45.84 (11.74) |
| Work in a practice, N (%) [*] | 220 (78.57%) | 146 (75.26%) | 74 (86.05%) |
| Imaging nearby, N (%) [*] | 114 (40.71%) | 77 (39.69%) | 37 (43.02%) |

[*] n = 280.

characteristics, we found significantly better agreement in the case of pharyngitis for USA physicians compared to Swiss physicians ($k_w$ = 0.76 vs. 0.61, p <0.001), and for physicians below 50-years-old compared to older physicians ($k_w$ = 0.76 vs. 0.57, p <0.001). See S3 Table for complete results.

When comparing the physician's decision with their estimated probability of disease, the agreement was higher for pharyngitis than for appendicitis ($k_w$ = 0.70, 95% CI 0.66–0.73 versus 0.61, 95% CI 0.56–0.66; Fig 1B and Table 2). Significantly better agreement was found in the case of pharyngitis for USA physicians than for Swiss physicians ($k_w$ = 0.81 vs. 0.64, p <0.001), under age 50 than for older physicians ($k_w$ = 0.75 vs. 0.64, p = 0.005), and for male patients than for female patients ($k_w$ = 0.74 vs. 0.65, p < 0.019) (S4 Table).

Probability of disease was correctly estimated as lying in the probability range of the corresponding vignette for 29% and 39% of physicians for appendicitis and pharyngitis, respectively. More than 60% of physicians underestimated the probability of appendicitis, while almost 40% overestimated probability for pharyngitis (Fig 1C and Table 3). Male physicians showed a higher percentage of underestimation and a lower percentage of overestimation with respect to female physicians (p = 0.042) (S5 Table).

Physicians started testing when the true probability was 52% for appendicitis and 16% for pharyngitis (test thresholds). They started initiating treatment when the true probability was 69% for appendicitis and 88% for pharyngitis (treatment thresholds) (Fig 2). A significantly higher test threshold was found for appendicitis for Swiss compared to USA physicians (54% vs. 48%, p = 0.006). The treatment threshold for appendicitis was higher for USA physicians

**Table 2. Concordance between physician decision (symptomatic, investigation or treatment) and true, respectively estimated probability of disease (low, medium or high).**

|  |  | **Appendicitis** | | | **Pharyngitis** | | |
|---|---|---|---|---|---|---|---|
|  |  | Symptomatic[a] | Investigation[b] | Treatment[c] | Symptomatic | Investigation | Treatment |
| **True probability** | Low | 167 | 20 | 1 | 172 | 18 | 1 |
|  | Medium | 38 | 94 | 63 | 46 | 134 | 8 |
|  | High | 5 | 57 | 129 | 7 | 127 | 61 |
|  | **Weighted kappa (95% CI)** | 0.74 (0.70–0.78) | | | 0.66 (0.62–0.71) | | |
|  |  | Symptomatic | Investigation | Treatment | Symptomatic | Investigation | Treatment |
| **Estimated probability** | Low | 194 | 97 | 27 | 186 | 14 | 0 |
|  | Medium | 3 | 54 | 80 | 23 | 92 | 0 |
|  | High | 1 | 20 | 83 | 5 | 171 | 70 |
|  | **Weighted kappa (95% CI)** | 0.61 (0.56–0.66) | | | 0.70 (0.66–0.73) | | |

[a]Symptomatic = symptomatic management only with no testing or treatment.

[b]Investigation = further investigation recommended.

[c]Treatment = treatment should be initiated.

**Table 3. Concordance between estimated and true probability.**

| | | APPENDICITIS | | | | | |
|---|---|---|---|---|---|---|---|
| | | Estimated probability % | | | | | |
| | | *0–39* | *47–58* | *50–64* | *57–71* | *61–79* | *66–99* |
| **True probability %** | *0–39* | **89** | 3 | 0 | 0 | 0 | 0 |
| | *47–58* | 87 | **0** | 0 | 0 | 0 | 0 |
| | *50–64* | 48 | 28 | **8** | 2 | 4 | 7 |
| | *57–71* | 42 | 25 | 7 | **3** | 10 | 8 |
| | *61–79* | 11 | 24 | 10 | 0 | **20** | 24 |
| | *66–99* | 12 | 16 | 9 | 2 | 19 | **41** |
| | | Underestimation | | Correct | | Overestimation | |
| | | 61% | | 29% | | 10% | |
| | | PHARYNGITIS | | | | | |
| | | Estimated probability % | | | | | |
| | | *1–2.5* | *5–10* | *11–17* | *28–35* | *51–53* | *54–100* |
| **True probability %** | *1–2.5* | **30** | 54 | 3 | 0 | 0 | 0 |
| | *5–10* | 6 | **63** | 10 | 4 | 3 | 2 |
| | *11–17* | 5 | 15 | **21** | 18 | 17 | 19 |
| | *28–35* | 4 | 11 | 14 | **22** | 28 | 10 |
| | *51–53* | 0 | 1 | 2 | 9 | **37** | 49 |
| | *54–100* | 0 | 1 | 3 | 9 | 33 | **48** |
| | | Underestimation | | Correct | | Overestimation | |
| | | 22% | | 39% | | 39% | |

than for Swiss physicians (74% vs. 68%, p = 0.008), for physicians less than 50-years-old compared to older physicians (71% vs 67%, p = 0.046) and when the patient was a woman than for male patients (71% vs. 68%, p = 0.048). It was also higher in the case of pharyngitis for Swiss physicians than for US physicians (100% vs. 61%, p < 0.001) and for physicians > 50 years old than for younger physicians (99% vs 78%, p = 0.005) (Table 4).

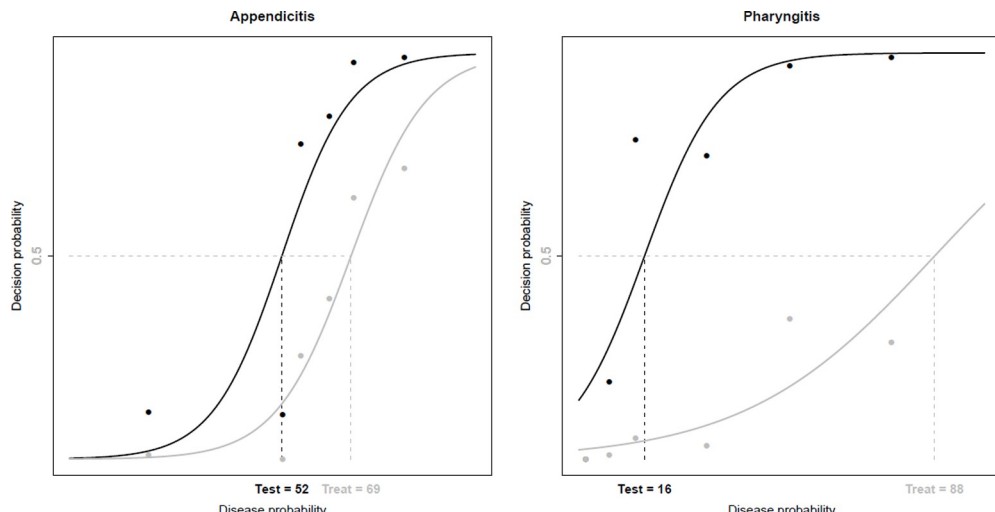

**Fig 2. Test and treatment thresholds for appendicitis and pharyngitis (obtained as equal to 0.5 the probability of not ruling out (test threshold) and treating (treatment threshold) estimated according to model (1).** Points represent empirical frequencies of decisions according to the true disease probability). Test: test threshold, Treat: treatment threshold.

**Table 4. Test and treatment thresholds for different subgroup populations.**

| | | | Appendicitis | | Pharyngitis | |
|---|---|---|---|---|---|---|
| | | | Estimate (95%CI) | p-value of the difference | Estimate (95%CI) | p-value of the difference |
| Test threshold | Country | CH | 54.1 (49.9–56.6) | **0.006** | 16.5 (13.9–19.5) | 0.708 |
| | | USA | 47.7 (40.6–52.2) | | 15.6 (11.7–20.0) | |
| | Sex physician | Male | 53.6 (49.1–56.2) | 0.051 | 16.3 (13.6–19.6) | 0.780 |
| | | Female | 49.4 (42.7–53.5) | | 15.7 (12.2–19.5) | |
| | Age physician | ≤ 50 | 53.1 (48.6–55.9) | 0.307 | 14.5 (11.8–17.6) | 0.121 |
| | | > 50 | 51.2 (45.2–54.6) | | 17.9 (14.6–22.0) | |
| | Sex patient | Male | 52.2 (46.8–55.4) | 0.899 | 17.6 (14.7–21.3) | 0.147 |
| | | Female | 52.4 (47.5–55.4) | | 14.5 (11.5–18.0) | |
| Treatment threshold | Country | CH | 67.5 (65.2–71.3) | **0.008** | 100 (70.4–100) | **<0.001** |
| | | USA | 73.7 (68.4–82.7) | | 61.1 (42.3–100) | |
| | Sex physician | Male | 68.0 (65.5–72.1) | 0.256 | 87.4 (58.6–100) | 0.931 |
| | | Female | 70.5 (66.5–77.0) | | 86.8 (58.6–100) | |
| | Age physician | ≤ 50 | 70.7 (67.4–76.6) | **0.046** | 78.1 (53.0–100) | **0.005** |
| | | > 50 | 66.8 (64.2–70.4) | | 99.3 (66.5–100) | |
| | Sex patient | Male | 67.6 (64.7–72.2) | **0.048** | 86.9 (59–100) | 0.832 |
| | | Female | 71.4 (67.7–78.1) | | 88.1 (59.6–100) | |

## Discussion

Only a minority of GPs correctly estimated the probability that a given patient had a disease based on clinical vignettes. Nevertheless, misestimating the probability of disease does not necessarily prevent GPs from choosing an appropriate clinical management decision according to recommendations. Indeed, this study revealed that their decision-making is highly congruent with proposed recommendations, independent of their (mis)estimation of the probability of disease, especially for appendicitis and to a lesser extent for pharyngitis.

Pharyngitis is commonly encountered in general practice. However, it is difficult to differentiate bacterial (often streptococcal) from non-bacterial infections. In this study, GPs tended to overestimate the risk of having a bacterial pharyngitis. Despite difficulty in estimating correctly probability, a majority of GPs gave symptomatic treatment when probabilities (true and estimated) were low and did a test when they were moderate, which is in line with recommendations. However, if the risk was high or if they estimated it as high, they still did a test rather than treating empirically with antibiotics, which is recommended by guidelines. This decision can be explained by different recommendations depending on the country or on local habits.

For appendicitis, GPs tended to globally underestimate the probability that the patient in the vignette had the disease. In patients with abdominal pain who have elevated inflammatory biomarkers in the blood, the differential diagnosis is broad. However, once again, bad estimation of the probability had little impact on the decision-making as the majority of clinical management decisions corresponded to the recommendations.

Recently, several studies have highlighted disparities in care management related to the sex of the patient, ranging from the choice of tests to establish the diagnosis to the treatment and clinical management decisions. For example, in the case of chest pain, women are less likely to be investigated (stress test, electrocardiogram or cardiac enzyme assay) even when data are adjusted to the fact that they may have an "atypical" clinical presentation [15]. This may partly explain their higher mortality after myocardial infarction [16]. In addition, in the case of abdominal pain, women are more likely to receive low-dose analgesics, or even anxiolytics, while men receive more opioids [17]. In the present study, GPs initiated treatment for

appendicitis faster (i.e. with a lower probability of disease) if the patient was male. Given the presentation of pain in the abdomen with some perturbation in blood parameters, GPs might lean towards diagnosis of appendicitis in males while for women they might first consider a gynecological problem. Despite the fact that the incidence of appendicitis is slightly higher in males, and especially in the pediatric age range [12, 18], it may also reflect a stereotyped belief of GPs that women overplay the situation.

On the other hand, studies have suggested that the gender of the caregiver also influences the future of the patient. For example, patients followed by female doctors have lower mortality and readmission rates than those followed by males [19]. Here, female GPs were better at estimating probabilities; their values were more often in the same risk category (range of probability) as the vignettes. However, it does not appear to have an effect because their clinical management choices are similar to male GPs (no significant difference).

Generally, the test threshold depends on the severity of the disease and therefore, it might be expected that appendicitis will be investigated with lower probability than pharyngitis, in view of potential complications. However, the test threshold was lower for pharyngitis than appendicitis (16% vs 52%, respectively) in this study. The availability of a rapid, inexpensive and specific test for streptococcal antigen is a likely reason. In the case of appendicitis, a blood test can be informative, but GPs have to organize an ultrasound or CT-scan to confirm the diagnosis, which may prevent some of them from doing so if the probability is not high. For appendicitis, GPs from the USA tend to start investigations with lower a probability than Swiss GPs. This could be a result of the risk of lawsuit, local culture, patient expectations for rapid diagnosis, reimbursement for testing or an easier access to imaging (US/CT) in the USA (43%, vs. 39% in Switzerland).

Looking at treatment thresholds brings to light differences in practice between the two countries. In Switzerland, our results reveal that Swiss GPs initiate treatment with antibiotics only when the probability of having a streptococcal pharyngitis is 100% (treatment threshold = 100) vs 61.1% in the USA. An explanation could be the impregnation of Swiss GPs who are trained with guidelines using the Centor score recommending a rapid test even if the probability of having a bacterial pharyngitis is high [20].

Finally, 78% of the responding physicians work in a private practice where laboratory tests are often limited, as are imaging capabilities. The remaining 22% are physicians (mostly in training), working in outpatient medical centers or other hospital-related public structures, often with a larger technical platform available (such as Unisanté, Lausanne). It is therefore conceivable that management differs according to the availability of diagnostic techniques and the degree of physicians' training.

### Implications for practice and score development

This study indicates that GPs are not skilled at accurately estimating disease probability based on signs, symptoms, and biomarkers. This means that they use other ways, perhaps more subjective, to make decisions, such as clinical judgment or clinical reasoning that integrates factors other than only the risk of disease. Clinical reasoning is defined as "the thinking and decision-making processes that enable a GPs to propose clinical management in a specific context of health problem solving". First, physicians will collect the information reported by the patient (anamnesis) and examine it (status) in order to look for clinical signs. Intuitively, they will synthesize all the information and then integrate it using their knowledge and experience (clinical sense) to develop a diagnostic hypothesis and propose a treatment. Most often unconsciously, they will estimate the probability that their patient has one or another of the suspected diagnoses. A complex process that integrates both epidemiological (prevalence) and clinical

presentation (signs and symptoms) concepts. More qualitatively, when physicians make a decision, they must be confident with it, not putting the patient too much at risk.

Physicians primarily use two types of clinical reasoning: "non-analytical" and "analytical". The first is an intuitive, quick and almost automatic reasoning that includes "pattern recognition" (where the clinician establishes a diagnosis following recognition of a characteristic pattern of signs and symptoms) and the "concrete cases" (where the clinician will remember a similar previously encountered case). The second type of reasoning includes the "hypothetico-deductive process", based on generation of hypotheses, which are constantly analyzed according to new information collected by physicians. However, factors may intervene that modify both clinical reasoning and diagnostic process. These may be related to the physician (age, gender, level of training/experience, fear of penal procedures, stereotypes, etc.), the patient (age, gender, manner of expressing symptoms, level of health literacy, etc.) or the context (private office vs hospital, degree of emergency, accessibility of diagnostic tests).

One interpretation is that the physician needs to make a binary decision: it is or it is not. Alternatively: is he/she sufficiently confident with this specific clinical presentation and his/her judgment to exclude the diagnosis and ultimately is his/her decision justifiable to him/her and to the patient. This is why medicine is also called "an art" and transforming all clinical management into a probabilistic approach is in vain.

In our particular context, probabilities might not have much importance in decision-making. A clinical decision is never neutral, but needs to be meaningful. In times where probabilities are used more and more to make decisions (considering genetic testing or oncology treatments for example), it seems important to explore other elements that influence choice.

## Limitations

The study has several limitations. First, very few studies have reported the probability of having the disease for each number of points of the score but often mention only broad categories (<4 = low risk,...).This approach by categories is very helpful to clinicians for making decisions [21], but not for the specific needs of the present study, which requires individual point-score probability estimates for given specific clinical patterns. This was a limit for creating the vignettes, especially for appendicitis [22–24]. Indeed, the probabilities used for appendicitis, especially with low scores are very high, probably higher than in other studies using aggregations of values of the Alvarado score. It is also well known that the setting modifies the probability (and the predictive value) of having a disease. For example, the probability of having an acute coronary disease in patients with chest pain is higher in the emergency room than in a primary care practice. The consequence is that the calculated test thresholds (based on probabilities provided by the retained studies, performed in hospitals) are doubtless too high. However, even if the absolute value might seem somehow unrealistic (almost 40% for an Alvarado score <2), the comparison of the decision of the GPs with the one provided by the score's interpretation is still valid. Indeed, the decision made by clinicians in the study is independent of the probability value. This might also explain why the estimation of the probabilities by the GPs are not very accurate (perhaps closer to the true probability in primary care). This might reinforce the idea that GPs make their decisions based on their experience and "gut feeling" rather than on probabilities.

Second, using clinical vignettes is perhaps subject to bias, as it does not allow reproducing situations of GPs with real patients in front of them. Probably the GPs relied mainly on the raw information present in the text of the vignette. The patient's perspective is not very present but must be considered through the patient's complaint. For example, a patient may say that he has a calf pain but the GP might not necessarily suspect and rule out deep vein thrombosis

after clarifying symptoms and performing clinical examination. Moreover, the influence of other factors such as the patients' attitude, or expectations, as well as the time available or the clinical sense of the GP, are difficult to reproduce. However, it is likely that factors such as the age, provenance or gender of GPs, and the patients' gender to a lesser extent, can still be explored by this approach.

Thirdly, the recruitment methods were different between the two countries: In Switzerland, all participants were contacted in person while in the USA GPs answered an advertisement on the Daily POEMs website. Due to this distinction, the two populations may be dissimilar, and a self-selection bias cannot be excluded. However, ruling out potential differences in imaging facilities, the characteristics of the GPs were similar, so it is unlikely that the two recruitment methods resulted in the inclusion of GPs with different profiles. Finally, clinical scores are associated with different types of recommendations mostly based on expert opinions. They differ according to places and to the evolution of practices. Those used in this study correspond to the recommendations commonly in place at the time of conceptualization. Indeed, the latest recommendations for streptococcal pharyngitis suggest that antibiotics should no longer be given systematically, even if a culture or rapid test is positive.

## Conclusions

This study brings to light the finding that general practitioners often misestimate the probability of disease, which might be due to important variations in probabilities depending on the setting. However, and paradoxically, their clinical management remains excellent, in line with recommendations for a given clinical presentation. This might suggest that numbers do not matter in decision-making, but rather the meaningfulness of the decision. Nevertheless, the possibility that over- or underestimation of risk may have an effect on a patient's prognosis should not be excluded. Further research should explore the elements that influence the decision-making process in the course of consultations, especially in respect of subjective factors that can affect judgment.

## Supporting information

**S1 Text. Example of a pharyngitis vignette (corresponding to a McIsaac score of 4).**
(DOCX)

**S1 Table. Risk intervals for appendicitis (Alvarado score) and pharyngitis (McIsaac score).**
(PDF)

**S2 Table. Management option suggested for pharyngitis and appendicitis.**
(PDF)

**S3 Table. Weighted kappa concordance between physician decision (symptomatic, investigation or treatment) and true probability of disease (low, medium or high) by patient and physician characteristics.**
(PDF)

**S4 Table. Weighted kappa concordance between physician decision (symptomatic, investigation or treatment) and estimated probability of disease (low, medium or high) by patient and physician characteristics.**
(PDF)

**S5 Table. Concordance between estimated and true probability by patient and physician characteristics.** [a]Underestimation, [b]Correct estimation, [c]Overestimation.
(PDF)

## Author Contributions

**Conceptualization:** Carole Clair, Mark H. Ebell, Nicolas Senn.

**Data curation:** Isabella Locatelli.

**Formal analysis:** Isabella Locatelli.

**Investigation:** Lionel De Alencastro.

**Methodology:** Nicolas Senn.

**Software:** Lionel De Alencastro.

**Supervision:** Carole Clair, Nicolas Senn.

**Validation:** Mark H. Ebell, Nicolas Senn.

**Writing – original draft:** Lionel De Alencastro.

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
