## [Decision Letter · Decision Letter 0]

28 Aug 2020

PONE-D-20-20579

Correlation of clinical decision-making with probability of disease: A web-based study among general practitioners

PLOS ONE

Dear Dr. De Alencastro,

Thank you for submitting your manuscript to PLOS ONE. After careful consideration, we feel that it has merit but does not fully meet PLOS ONE’s publication criteria as it currently stands. Therefore, we invite you to submit a revised version of the manuscript that addresses the points raised during the review process.

We look forward to receiving your revised manuscript.

Kind regards,

Itamar Ashkenazi

Academic Editor

PLOS ONE

Journal Requirements:

Reviewers' comments:

Reviewer's Responses to Questions

**Comments to the Author**

1. Is the manuscript technically sound, and do the data support the conclusions?

Reviewer #1: Yes

Reviewer #2: Yes

Reviewer #3: Yes

2. Has the statistical analysis been performed appropriately and rigorously? 

Reviewer #1: Yes

Reviewer #2: Yes

Reviewer #3: Yes

3. Have the authors made all data underlying the findings in their manuscript fully available?

Reviewer #1: Yes

Reviewer #2: Yes

Reviewer #3: Yes

4. Is the manuscript presented in an intelligible fashion and written in standard English?

Reviewer #1: Yes

Reviewer #2: Yes

Reviewer #3: Yes

5. Review Comments to the Author

Reviewer #1: This is a very well designed study, covering GP's decision making when patients present a set of symptoms and markers of appendicitis and pharyngitis. My comments deal with the broader context in which this two-country study was conducted.

It seems well known that humans are remarkably poor at guiding their decisions by probability. Erratical behaviour, in spite of clear logical evidence against is a defining characteristic of our species. We don't function as machines and we are notoriously bad at understanding probabilities. Witness the difficulties students have in traditional math class to grasp the concept. It is thus no surprise, that GPs, tend to use their gut feeling and intuition in medical decision making. This makes sense. In their clinical reality, they use the tried and true methods to establish a diagnostic, and their experience cannot betray them. Given this, I have the following issues which I suggest the authors address in the paper.

1) In the guidelines for GP practice, there are often probability indicators associated with a given set of data. Do GPs use these probabilities in their clinical decision making ? Is this the basic issue the authors attempt to get at ? This is what the study stated aims hint at (line 95 to 100). Why would it be important and how would it help to gain greater understanding to : explore whether decisions in practice are in-line with the theoretical recommendations and to explore potential factors that can also influence clinical judgment ? The basic "who cares" question about this study needs to be better presented.

2) It appears incongruous to explore other factors that could influence GP decision-making by using pre-set vignettes that, by definition, greatly reduce the complexity of the situation. Would these vignettes be sufficiently "authentic" as to allow for observation of the influence of "other factors" in the clinical reasoning ? The aims of the paper need to be reformulated in order to be congruous with the experimental situation that is presented. In my view, aims should involve the verification or the confirmation of a hypothesis whereby GP's do or do not rely on probability recommendations to make clinical decisions.

3) Finally, it is unclear to me why Switzerland (Canton de Vaud) and the USA are the chosen countries. A more compelling argument should be made to explain why comparing a country of 8 million people with another of 250 million makes sense. What is the underlying connection? Do they use the same probability scores (Alvarado and McIsaac) ? And if they do, why bother to compare them ? When General Practice is reduced to 12 (line 116) or 24 (as stated in the abstract) standard vignettes, why should it matter the context where these practitioners ply their trade ? Same applies to the gender distinctions. How can approach with vignettes shed light on this issue beyond simply comparing the resulting scores.

Reviewer #2: Overall recommendation - good paper and interesting read. Easy to follow and understand. May want to consider adding the patient perspective when considering medical diagnosis. When considering GPS and looking at probability of disease and management decisions should be discussed in further detail. You mention clinical judgment or reasoning integrated factors other than disease, would like to have more detail surrounding what this entails.

Reviewer #3: Thank you for the opportunity to review the manuscript titled “Correlation of clinical decision-making with probability of disease: A web-based study among general practitioners.” The authors’ research addresses and interesting point and gives insight into how physicians make decisions. The manuscript is well written and informative. Please find my comments below.

The authors state they chose streptococcal pharyngitis and appendicitis partly because they are common clinical problems. Please provide information on how common these conditions are seen and treated by general practitioners. It may also be helpful to provide the common signs and symptoms for patient presentation.

Under statistical methods provide age ranges used for stratification.

How did the authors account for how long the participants had been in practice in the analysis?

The authors state that the eligibility criteria included working in a private practice or in a similar structure. What does that mean for the 22% not in private practice? What does similar structure refer to? Could there be a bias for those that were not in private practice?

The authors touch on it in the discussion section, but do the treatment guidelines differ significantly between the countries for these conditions? Additional details should be added to the discussion.

Under the limitations, the authors should address the potential for biases due to differences in recruitment methods between physicians from Switzerland versus the USA. There is likely a self-selection bias.

I recommend reviewing the manuscript for minor grammatical mistakes. There are several typos and a few places where the subject/verb agreement is not correct.

6. PLOS authors have the option to publish the peer review history of their article (what does this mean?). If published, this will include your full peer review and any attached files.

Reviewer #1: **Yes: **Nicolas Fernandez

Reviewer #2: No

Reviewer #3: No

---

## [Author Response · Author response to Decision Letter 0]

7 Oct 2020

I don't really undestand this step. All my comments and answers to reviewers are in the files "Response to Reviewers".

Please tell me if I do wrong.

---

## [Editor Report · Decision Letter 1]

12 Oct 2020

Correlation of clinical decision-making with probability of disease: A web-based study among general practitioners

PONE-D-20-20579R1

Dear Dr. De Alencastro,

We’re pleased to inform you that your manuscript has been judged scientifically suitable for publication and will be formally accepted for publication once it meets all outstanding technical requirements.

Kind regards,

Itamar Ashkenazi

Academic Editor

PLOS ONE
---

## [Editor Report · Acceptance letter]

19 Oct 2020

PONE-D-20-20579R1 

Correlation of clinical decision-making with probability of disease: A web-based study among general practitioners 

Dear Dr. De Alencastro:

I'm pleased to inform you that your manuscript has been deemed suitable for publication in PLOS ONE. Congratulations! Your manuscript is now with our production department. 

Kind regards, 

on behalf of

Dr. Itamar Ashkenazi 

Academic Editor

PLOS ONE